# Language-Conditioned Imitation Learning for Robot Manipulation Tasks

**Simon Stepputtis** [1]    **Joseph Campbell**[1]    **Mariano Phielipp**[2]
**Stefan Lee**[3]    **Chitta Baral**[1]    **Heni Ben Amor**[1]
[1]Arizona State University, [2]Intel AI Labs, [3]Oregon State University
{sstepput, jacampb1, chitta, hbenamor}@asu.edu
mariano.j.phielipp@intel.com    leestef@oregonstate.edu

## Abstract

Imitation learning is a popular approach for teaching motor skills to robots. However, most approaches focus on extracting policy parameters from execution traces alone (i.e., motion trajectories and perceptual data). No adequate communication channel exists between the human expert and the robot to describe critical aspects of the task, such as the properties of the target object or the intended shape of the motion. Motivated by insights into the human teaching process, we introduce a method for incorporating unstructured natural language into imitation learning. At training time, the expert can provide demonstrations along with verbal descriptions in order to describe the underlying intent (e.g., "go to the large green bowl"). The training process then interrelates these two modalities to encode the correlations between language, perception, and motion. The resulting language-conditioned visuomotor policies can be conditioned at runtime on new human commands and instructions, which allows for more fine-grained control over the trained policies while also reducing situational ambiguity. We demonstrate in a set of simulation experiments how our approach can learn language-conditioned manipulation policies for a seven-degree-of-freedom robot arm and compare the results to a variety of alternative methods.

## 1   Introduction

Learning robot control policies by imitation [31] is an appealing approach to skill acquisition and has been successfully applied to several tasks, including locomotion, grasping, and even table tennis [8, 2, 25]. In this paradigm, expert demonstrations of robot motion are first recorded via kinesthetic teaching, teleoperation, or other input modalities. These demonstrations are then used to derive a control policy that generalizes the observed behavior to a larger set of scenarios that allow for responses to perceptual stimuli (e.g., joint angles and an RGBD camera image of the work environment) with appropriate actions (e.g., moving a table-tennis paddle to hit an incoming ball).

In goal-conditioned tasks, perceptual inputs alone may be insufficient to dictate optimal actions [10] (e.g., without a target object, what should a picking robot retrieve from a bin when activated?). Consequently, expert demonstration and control policies must also be conditioned on a representation of the goal. While we use the term *goals*, it may refer to end goals (e.g., target objects) or constraints on motion (e.g., minimizing end-point effector acceleration) [14]. Prior work has typically employed manually designed goal specifications (e.g., vectors indicating a target position, a one-hot vector indicating target objects, or a single value indicating the execution speed). However, this is an inflexible approach that must be pre-defined before training and cannot be modified after deployment.

In the present work, we consider language as a flexible goal specification for imitation learning in manipulation tasks. As shown in Fig. 1(center), we consider a seven-degree-of-freedom robot

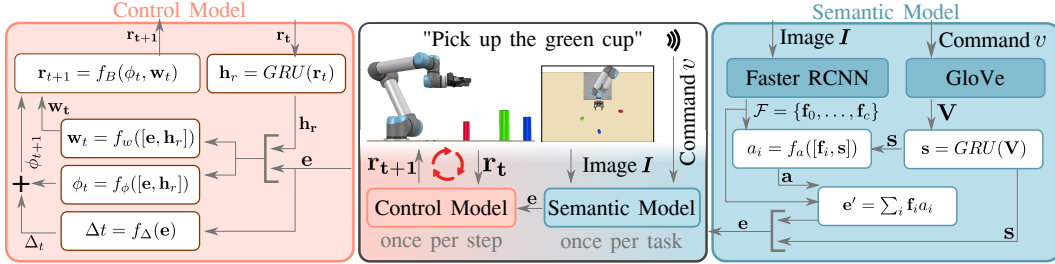

Figure 1: Overview of the general system architecture. (Left) Details of the controller model, which synthesizes robot control signals . (Right) details of the semantic model, which extracts critical information about the task from both perceptual input and language commands. Dark-blue boxes indicate pre-trained components of our model.

manipulator anchored to a flat workspace populated with a set of objects that vary in shape, size, and color. The agent is instructed by a user to manipulate these objects in language for picking (e.g., "grab the blue cup") and pouring tasks (e.g., "pour some of its contents into the small red bowl"). In order to succeed, the agent must relate these instructions to the objects in the environment, as well as constraints on how they are manipulated (e.g., pouring some or all of something require different motions). We examine the role of imitation learning from demonstrations in this setting that consist of developing a training set of instructions and associated robot motion trajectories.

We developed an end-to-end model for the language-conditioned control of an articulated robotic arm – mapping directly from observation pixels and language-specified goals to motor control. We conceptually divided our architecture into two modules: a high-level semantic network that encodes goals and the world state and a lower-level controller network that uses the higher encoding to generate suitable control policies. Our high-level semantic network must relate language-specified goals, visual observation of the work environment, and the robot's current joint positions into a single encoding. To do this, we leveraged advances in attention mechanisms from vision-and-language research [3] to associate instructions and target objects. Our low-level controller synthesizes parameters of a motor primitive that specifies the entire future motion trajectory, providing insight into the predicted future behavior of the robot from the current observation. The model was trained end-to-end to reproduce demonstrated behavior while minimizing a set of auxiliary losses to guide the intermediate outputs.

We evaluated our model in a dynamic-enabled simulator with random assortments of objects and procedurally generated instructions, with success in 84% of sequential tasks that required picking up a cup and pouring its contents into another vessel. This result significantly outperformed state-of-the-art baselines. We provided detailed ablations of modeling decisions and auxiliary losses, as well as detailed analysis of our model's generalization to combinations of modifiers (color, shape, size, and pour-quantity specifiers). We also assessed robustness to visual and physical perturbations in the environments. While our model was trained on synthetic language, we also ran human-user experiments with free-form natural-language instructions for picking/pouring tasks, with a success rate of 64% for these instructions.

All data used in this paper, along with a trained model and the full source code can be found at: `https://github.com/ir-lab/LanguagePolicies`. The release features a number of videos and examples on how to train and validate language-conditioned control policies in a physics-based simulation environment. Additionally, detailed information about the experimental setup and the human data collection process can be found under the link above.

**Contributions.** To summarize our contributions, we

– introduced a language-conditioned manipulation task setting in a dynamically accurate simulator,
– provided a natural-language interface which allows laymen users to provide robot task specifications in an intuitive fashion,
– developed an end-to-end, language-conditioned control policy for manipulation tasks composed of a high-level semantic module and low-level controller, integrating language, vision, and control within a single framework,
– demonstrated that our model, trained with imitation learning, achieved a high success rate on both synthetic instructions and unstructured human instructions.

## 2    Background

Imitation learning (IL) provides an easy and engaging way to teach new skills to an agent. Instead of programming, the human can provide a set of demonstrations [6] that are turned into functional [16] or probabilistic [23, 7] representations. However, a limitation of this approach is that the state representation must be carefully designed to ensure that all necessary information for adaptation is available. Furthermore, it is assumed that either a sufficiently large task taxonomy or set of motion primitives is already available (i.e., semantics and motions are not trained in conjunction). Neural approaches scale imitation learning [27, 4, 20, 1, 9] to high-dimensional spaces by enabling agents to learn task-specific feature representations. However, both foundational references [27], as well as more recent literature [10], have noted that these methods lack "a communication channel," which would allow the user to provide further information about the intended task, at nearly no additional cost [11]. Hence, both the designer (programmer) and the user have to resort to numerical approaches for defining goals. For example, a one-hot vector may indicate which of the objects on the table is to be grasped. This limitation results in an unintuitive and potentially hard-to-interpret communication channel that may not be expressive enough to capture user intent regarding *which* object to act upon or *how* to perform the task. Another popular methodology for providing such semantic information is to use formal specification languages such as temporal logic [19, 28]. Such formal frameworks are compelling, since they support the formal verification of provided commands. Even for experts, specifying instructions in these languages can be a challenging, complicated, and time-consuming endeavor. An interesting compromise was proposed in [15], where natural-language specifications were first translated to temporal logic via a deep neural network. However, such a methodology limits the range of descriptions that can be provided due to the larger expressivity of the English language relative to the formal specification language. DeepRRT, presented in [20], describes a path-planning algorithm that uses natural-language commands to steer search processes, and [32] introduced the use of language commands for low-level robot control. A survey of natural language for robotic task specification can be found in Matuszek [24]. Beyond robotics, the combination of vision and language has received ample attention in visual question-and-answering systems (VQA) [22, 5] and vision-and-language navigation (VNL) [34, 18, 10]. Our approach is most similar to [1]. However, unlike our model, the work in [1] used a fixed alphabet and required information about the task to be extracted from the sentence before being used for control. In contrast, our model can extract a variety of information directly from natural language.

## 3    Problem formulation and approach

We considered the problem of learning a policy $\pi$ from a given set of demonstrations $\mathcal{D} = \{\mathbf{d}^0, .., \mathbf{d}^m\}$, where each demonstration contained the desired trajectory given by robot states $\boldsymbol{R} \in \mathbb{R}^{T \times N}$ over $T$ time steps and with $N$ control variables[1]. We also assumed that each demonstration contained perceptual data $\boldsymbol{I}$ of the agent's surroundings and a task description $v$ in natural language. Given these data sources, our overall objective was to learn a policy $\pi(v, \boldsymbol{I})$, which imitated the demonstrated behavior in $\mathcal{D}$ while considering the semantics of the natural-language instructions and critical visual features of each demonstration. After training, we provided the policy with a different, new state for the agent's environment, given as image $\boldsymbol{I}$, and a new task description (instruction) $v$. In turn, the policy generated control signals that were needed to achieve the objective described in the task description. We did not assume any manual separation or segmentation into different tasks or behaviors. Instead, the model was assumed to independently learn such a distinction form the provided natural-language description. Fig. 1 shows an overview of our proposed method. At a high level, our model takes an image $\boldsymbol{I}$ and task description $v$ as input to create a task embedding $\boldsymbol{e}$ in the semantic model. Subsequently, this embedding is used in the control model to generate robot actions at each time in a closed-loop fashion.

### 3.1    Preprocessing vision and language

We first preprocessed both the input image and verbal description by building upon existing frameworks for image processing and language embedding. More specifically, we used a pre-trained object detection network on image $\boldsymbol{I} \in \mathbb{R}^{569 \times 320 \times 3}$ of the robot's environment that identified salient image

regions of any object found in the robot's immediate workspace. In our approach, we used Faster R-CNN [29] to identify a set of candidate objects $\mathcal{F} = \{\boldsymbol{f}_0, .., \boldsymbol{f}_c\}$, each represented by a feature vector $\boldsymbol{f} = [f^o, \boldsymbol{f}^b]$ composed of the detected class $f^o$, as well as their bounding boxes $\boldsymbol{f}^b \in \mathbb{R}^4$, within the workspace of the robot, ordered by the detection confidence $f^c$ of each class. Based on a pre-trained FRCNN model trained from ResNet-101 on the COCO dataset, we continued to fine-tune the model for our specific use-case on 40 thousand arbitrarily generated environments from our simulator. After fine-tuning, the certainty of FRCNN on our objects was above 98%.

Regarding the preparation of the language input, each verbal description $v$ was split into individual words and converted into a row index of a matrix $\boldsymbol{G} \in \mathbb{R}^{30000 \times 50}$, representing the 30 thousand most-used English words, initialized and frozen to utilize pre-trained GloVe word embeddings [26]. Our model took the vector of row-indices as input, and the conversion to their respective 50-dimensional word embedding was done within our model to allow further flexibility for potentially untrained words.

## 3.2   Semantic model

The goal of the semantic model is to identify relevant objects described in the natural-language command, given a set of candidate objects. In order to capture the information represented in the natural-language command, we first converted the task description $v$ into a fixed-size matrix $\boldsymbol{V} \in \mathbb{R}^{15 \times 50}$, encoding up to 15 words with their respective 50-dimensional word embedding. Based on $\boldsymbol{V}$, a sentence embedding $\boldsymbol{s} \in \mathbb{R}^{32}$ was generated with a single GRU cell $\boldsymbol{s} = \text{GRU}(\boldsymbol{V})$.

To identify the object referred to by the natural-language command $v$ from the set of candidate regions $\mathcal{F}$, we calculated a likelihood for each region given the sentence embedding $\boldsymbol{s}$ [3]. The likelihood $a_i = \boldsymbol{w}_a^T f_a([\boldsymbol{f}_i, \boldsymbol{s}])$ was calculated by concatenating the sentence embedding $\boldsymbol{s}$ with each candidate object $\boldsymbol{f}_i$, applying the attention network $f_a : \mathbb{R}^{37} \rightarrow \mathbb{R}^{64}$ and converting the result into a scalar by multiplying it with a trainable weight $\boldsymbol{w}_a \in \mathbb{R}^{64}$. The function $f_a(\boldsymbol{x}) = \tanh(\boldsymbol{W}\boldsymbol{x} + \boldsymbol{b}) \odot \sigma(\boldsymbol{W}'\boldsymbol{x} + \boldsymbol{b}')$ is a nonlinear transformation that used a gated hyperbolic tangent activation [12], where $\odot$ represents the Hadamard product of the elements, and $\boldsymbol{W}, \boldsymbol{W}', \boldsymbol{b}, \boldsymbol{b}'$ are trainable weights and biases, respectively. This operation was repeated for all $c$ candidate regions, and the individual likelihoods $a_i$ were used to form a probability distribution over candidate objects $\boldsymbol{a} = \text{softmax}([a_0, ..., a_c])$. Then, the language-conditioned task representation was the mean $\boldsymbol{e}' = \sum_{i=0}^{c} \boldsymbol{f}_i a_i$ where $\boldsymbol{e}' \in \mathbb{R}^5$. The final task representation $\boldsymbol{e} \in \mathbb{R}^{32}$ was computed by reintroducing the sentence embedding $\boldsymbol{s}$, which was needed in the low-level controller to determine task modifiers like *everything* or *some*, and concatenating it with $\boldsymbol{e}'$. The task embedding was then created with a single fully connected layer $\boldsymbol{e} = \text{ReLU}(\boldsymbol{W}[\boldsymbol{e}', \boldsymbol{s}] + \boldsymbol{b})$, where $\boldsymbol{W}$ and $\boldsymbol{b}$ were trainable variables.

## 3.3   Control model

The generation of controls is a function that maps a task embedding $\boldsymbol{e}$ and the current agent state $\boldsymbol{r}_t$ to control values for future time steps. Control signal generation is performed in two steps. In the first step, the control model produces the parameters that fully specify a motor primitive. A motor primitive in this context is a trajectory of the control signals for all the robot's degrees of freedom and can be executed in an open-loop fashion until task completion. However, to account for the nondeterministic nature of control tasks (e.g., physical perturbations, sensor noise, execution noise, force exchange, etc.) we employed a closed-loop approach by recalculating the motor primitive parameters at each time step.

**Motor primitive generation** We used motor primitives inspired by the approach in [16]. A motor primitive was parameterized by $\boldsymbol{w} \in \mathbb{R}^{1 \times (B*7)}$, where $B$ is the number of kernels for each DOF of the robot. These parameters specified the weights for a set of radial basis function (RBF) kernels, which would be used to synthesize control signal trajectories in space. In addition, the motor primitive generation step also estimated the current (temporal) progress towards the goal as phase variable $0 \leq \phi \leq 1$ and the desired phase progression $\Delta_\phi$. A phase variable of $0$ means that the behavior has not yet started, while a value of $1$ indicates a completed execution. Predicting the phase and phase progression allowed our model to dynamically update the speed at which the policy was evaluated. In order to keep track of the robot's current and previous movements, we used a GRU cell that was initialized with the start configuration $\boldsymbol{r}_0$ of the robot and encoded all subsequent robot states $\boldsymbol{r}_t$ at each step $t$ of the control loop into a latent robot state $\boldsymbol{h}_t \in \mathbb{R}^7$. Based

on the task encoded in the latent task embedding $e$ and the latent state of the robot $h_t$ at time step $t$, the model generated the full set of motor primitive parameters for the current time step $(w_t, \phi_t, \Delta_\phi) = (f_w([h_t, e]), f_\phi([h_t, e]), f_\Delta(e))$, where $f_\phi : \mathbb{R}^{39} \to \mathbb{R}^1$, $f_w : \mathbb{R}^{39} \to \mathbb{R}^{B*7}$ and $f_\Delta : \mathbb{R}^{32} \to \mathbb{R}^1$ are multilayer perceptrons. Finally, the generated parameters were used to synthesize and execute robot control signals, as described in the next section.

**Motor primitive execution** A motor primitive parameterization $(w_t, \phi_t, \Delta_\phi)$ encoded the full trajectory for all robot DOF. To generate the next control signals $r_{t+1}$ to be executed, we evaluated the motor primitive at phase $\phi_t + \Delta_\phi$. Each motor primitive was a weighted combination of radial basis function (RBF) kernels positioned equidistantly between phases 0 and 1. Each kernel was characterized by its location $\mu$ with a fixed scale $\sigma$: $\Phi_{\mu,\sigma}(x) = \exp\left(-\left((x - \mu)^2 \big/ (2\sigma)\right)^2\right)$. All $B$ kernels for a single DOF were represented as a basis function vector $[\Phi\mu_0, \sigma(\phi), \dots, \Phi\mu_B, \sigma(\phi)] \in \mathbb{R}^{B \times 1}$, and each kernel was a function of $\phi$, representing the relative temporal progress towards the goal. Given that a linear combination of RBF kernels approximated the desired control trajectory, we could define a sparse linear map $H_{\phi_t} \in \mathbb{R}^{7 \times (B*7)}$, which contained the basis function vectors for each DOF along the diagonals. The control signal at time $t+1$ was given as $r_{t+1} = f_B(\phi_t + \Delta_\phi, w_t) = H_{\phi_t + \Delta_\phi} w_t^T$, which allowed us to quickly calculate the target joint configuration at a desired phase $\phi_t + \Delta_\phi$ in a single multiplication operation. The respective parameters were generated in the previously described motor primitive generation step. The control model worked in a closed loop, taking potential discrepancies (and perturbations) between the desired robot motion and the actual motion of the robot into consideration. Based on the past motion history of the robot, our model was able to identify its progress within the desired task by utilizing phase estimation. This phase estimation was a unique feature in our controllers and differed from previous approaches with a fixed phase progression [16].

## 3.4 Model integration

The components described in the previous sections were combined sequentially to create our final model. After preprocessing the input command into a sequence of word IDs for GloVe and detecting object locations in the robot's immediate surrounding using FRCNN, the semantic model (section 3.2) created a task-specific embedding $e$ that encoded all the necessary information about the desired action. Subsequently, the control model translated the latent task embedding $e$ and current robot state $r_t$ at each time step $t$ into hyper-parameters for a motor primitive (section 3.3). These parameters were defined as the weights $w_t$, phase $\phi_t$, and phase progression $\Delta_\phi$ at time $t$. By using these parameters, the motor primitive was used to infer the next robot configuration $r_{t+1}$, as well as the entire trajectory $R = \{r_0, ..., r_T\}$, allowing for subsequent motion analysis. At each time step, a new motor primitive was defined by generating a new set of hyper-parameters from the task representation $e$. While $e$ was constant over the duration of an interaction, the current robot state $r_t$ was used at each time step to update the motor primitive's parameters. An overview of the architecture can be seen in figure 1.
The integration of our model resulted in an end-to-end approach that takes high-level features and directly converts them to low-level robot control parameters. As opposed to a multi-staged approach, which requires a significant amount of additional feature engineering, our framework learned how language affects the behavior (type, goal position, velocity, etc.) automatically, while also learning the control itself. Another advantage of this end-to-end approach was that the overall system could be trained such that the individual components harmonized. This was particularly important for the interplay of language embedding and control when using language as a modifier for trajectory behaviors.

## 3.5 End-to-end training

Our model was trained in an end-to-end fashion, utilizing five auxiliary losses to aid the training process. The overall loss was a weighted sum of five auxiliary losses: $\mathcal{L} = \alpha_a \mathcal{L}_a + \alpha_t \mathcal{L}_t + \alpha_\phi \mathcal{L}_\phi + \alpha_w \mathcal{L}_w + \alpha_\Delta \mathcal{L}_\Delta$. The guided attention loss $\mathcal{L}_a = -\sum_i^C x_i \log(y_i)$ trained the attention model and was defined as the cross-entropy loss for a multi-class classification problem over $c$ classes, where $a^\star \in \mathbb{R}^c$ are the ground truth labels and $a \in \mathbb{R}^c$ the predicted classes. The training label $a^\star$ was a one-hot vector created alongside the image preprocessing. It indicated which object is referred

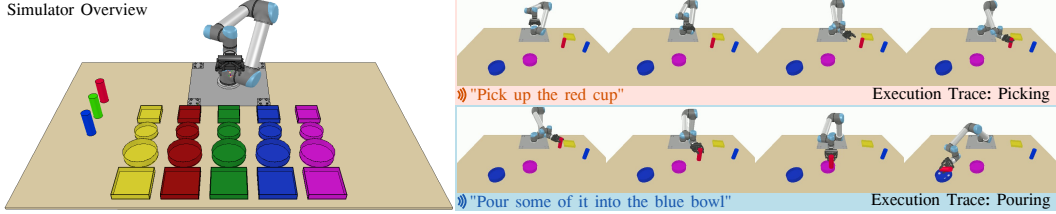

Figure 2: Overview of the available objects in simulation (left) and sample task execution sequences with their respective commands of the two tasks: picking (top right) and pouring (bottom right).

to by the task description, depending on the order of candidate objects in $\mathcal{F}$. The controller was guided by four mean-squared-error losses, starting with the phase estimation $\mathcal{L}_\phi = \mathrm{MSE}(\phi_t, \phi_t^\star)$ and with the phase progression, defined as $\mathcal{L}_\Delta = \mathrm{MSE}(\Delta_\phi, \Delta_\phi^\star)$, indicating where the robot was in its current behavior and how much the current configuration would be updated for the next time steps. Both of the labels $\Delta_\phi^\star$ and $\phi_t^\star$ could easily be inferred from the number of steps in the given demonstration. Furthermore, we minimized the difference between two consecutive basis weights with $\mathcal{L}_w = \mathrm{MSE}(\boldsymbol{W}_t, \boldsymbol{W}_{t+1})$. By minimizing this loss, the model was ultimately able to predict full motion trajectories at each time step, since significant updates between consecutive steps were mitigated. Finally, the overall error of the generated trajectory $\boldsymbol{R} = [\boldsymbol{r}_{\phi=0}, ..., \boldsymbol{r}_{\phi=1}]$ was calculated via $\mathcal{L}_t = \mathrm{MSE}(\boldsymbol{R}, \boldsymbol{R}^\star)$ against the demonstrated trajectory $\boldsymbol{R}^\star$. Values $\alpha_a = 1$, $\alpha_t = 5$, $\alpha_\phi = 1$, $\alpha_w = 50$, $\alpha_\Delta = 14$ were empirically chosen as hyper-parameters for $\mathcal{L}$ that had been found by a grid-search approach. We trained our model in a supervised fashion by minimizing $\mathcal{L}$ with an Adam optimizer using a learning rate of $0.0001$.

# 4    Evaluation and results

We evaluated our approach in a simulated robot task with a table-top setup. In this task, a seven-DOF robot manipulator had to be taught by an expert how to perform a combination of picking and pouring behaviors. At training time, the expert provided both a kinesthetic demonstration of the task and a verbal description (e.g.,"pour a little into the red bowl"). The table might feature several differently shaped, sized, and colored objects, which often led to ambiguities in natural-language descriptions thereof. The robot had to learn how to efficiently extract critical information from the available raw-data sources in order to determine *what* to do, *how* to do it, or *where* to go. We show that our approach leveraged perception, language, and motion modalities to generalize the demonstrated behavior to new user commands or experimental setups.

The evaluation was performed using CoppeliaSim [30, 17], which allowed for accurate dynamics simulations at an update rate of 20Hz. Fig. 2 depicts the table-top setup and the different variations of the objects used. We utilized three differently colored cups containing a granular material that could be poured into the bowls. Additionally, we used 20 variations of bowls in two sizes (large and small), two shape types (round and squared), as well as five colors (red, green, blue, yellow, and pink). When generating an environment, we randomly placed a subset of the objects on the table, with a constraint to prevent collisions or other artifacts. A successful picking action was achieved when a grasped object could be stably lifted from the table. Successful pouring was detected whenever the cup's dispersed content ended up in the correct bowl. Tasks of various difficulties could be created by placing multiple objects with overlapping properties on the table.

To generate training and test data, we asked five human experts to provide templates for verbal task descriptions. Annotators watched prerecorded videos of a robot executing either of the two tasks (picking or pouring) and were asked to issue a command that they thought the robot was executing. During annotation, participants were encouraged to use free-form natural language and not adhere to a predefined language pattern. The participants in our data collection were graduate students familiar with robotics but not familiar with the goal of the present research. Overall, we collected 200 task explanations from five annotators where each participant labeled 20 picking and 20 pouring actions. These 200 task descriptions were then manually templated to create replaceable noun phrases and adverbs, as well as basic sentence structures. To train our model, task descriptions for the training examples were then automatically generated from the set of sentence templates and synonyms from

Table 1: Model ablations concerning auxiliary losses, model structure, and dataset size.

| | Our model | | | | | Tasks Success | | | Execution statistics | | | | | Error statistics (pouring) | | | | | | | |
|---|---|---|---|---|---|---|---|---|---|---|---|---|---|---|---|---|---|---|---|---|---|
| | Att | $\Delta_\phi$ | $\phi$ | $W$ | Trj | Pick | Pour | Seq | Dtc | PIn | QDif | MAE | Dst | None | C | S | F | C+S | C+F | S+F | C+S+F |
| 1 | | | | | ✓ | 0.57 | 0.53 | 0.28 | 0.83 | 0.61 | 0.79 | 0.15 | 9.33 | 0.83 | 0.36 | 0.69 | **1.00** | 0.31 | 0.00 | 0.90 | 0.56 |
| 2 | | ✓ | ✓ | ✓ | ✓ | 0.00 | 0.44 | 0.00 | 0.67 | 0.57 | 0.74 | 0.17 | 20.78 | **1.00** | 0.33 | 0.62 | 0.50 | 0.27 | 0.00 | 0.80 | 0.3 |
| 3 | ✓ | | | | ✓ | 0.62 | 0.84 | 0.51 | 0.97 | 0.89 | 0.94 | 0.12 | 4.16 | 0.83 | **0.89** | 0.85 | **1.00** | 0.82 | 0.67 | 0.80 | 0.67 |
| 4 | FF attention | | | | | 0.00 | 0.01 | 0.00 | 0.41 | 0.14 | 0.60 | 0.22 | 25.63 | 0.00 | 0.00 | 0.00 | 0.00 | 0.07 | 0.00 | 0.00 | 0.00 |
| 5 | RNN controller | | | | | 0.02 | 0.00 | 0.00 | 0.44 | 0.17 | 0.71 | 0.38 | 19.72 | 0.00 | 0.00 | 0.00 | 0.00 | 0.00 | 0.00 | 0.00 | 0.00 |
| 6 | FF step orediction | | | | | 0.91 | **0.87** | 0.79 | **0.96** | 0.93 | **0.96** | 0.06 | 4.41 | **1.00** | 0.86 | 0.87 | 0.88 | 0.82 | 0.67 | **1.00** | 0.89 |
| 7 | Dataset size 2,500 | | | | | 0.69 | 0.15 | 0.10 | 0.67 | 0.36 | 0.55 | 0.18 | 13.92 | 0.33 | 0.06 | 0.23 | 0.50 | 0.00 | 0.00 | 0.50 | 0.00 |
| 8 | Dataset size 5,000 | | | | | 0.58 | 0.17 | 0.10 | 0.69 | 0.39 | 0.65 | 0.20 | 11.57 | 0.67 | 0.09 | 0.15 | 0.67 | 0.08 | 0.00 | 0.30 | 0.0 |
| 9 | Dataset size 10,000 | | | | | 0.54 | 0.55 | 0.29 | 0.86 | 0.65 | 0.67 | 0.11 | 7.17 | 0.83 | 0.42 | 0.69 | **1.00** | 0.35 | 0.33 | 0.90 | 0.44 |
| 10 | Dataset size 20,000 | | | | | 0.80 | 0.72 | 0.59 | 0.90 | 0.84 | 0.91 | 0.13 | 8.81 | 0.83 | 0.71 | 0.85 | **1.00** | 0.71 | 0.33 | 0.70 | 0.56 |
| 11 | Dataset size 30,000 | | | | | 0.94 | 0.86 | 0.80 | 0.94 | **0.95** | 0.94 | **0.05** | **4.12** | 0.67 | 0.86 | **0.92** | **1.00** | **0.88** | 0.33 | 0.90 | **1.00** |
| 12 | **Our model** | | | | | **0.98** | 0.85 | **0.84** | 0.94 | 0.94 | 0.94 | **0.05** | 4.85 | 0.83 | 0.83 | 0.85 | **1.00** | **0.88** | **1.00** | 0.70 | 0.89 |

which multiple sentences could be extracted via synonym replacement. In order to generate natural task descriptions, we first identified the minimal set of visual features required to uniquely identify the target object, breaking ties randomly. The set of required features was dependent on which objects were in the scene – e.g., if only one red object existed, a viable description that uniquely describes the object in the given scene could refer only to the target's color; however, if multiple red objects were present, other or further descriptors might be necessary. Synonyms for objects, visual feature descriptors, and verbs were chosen at random and applied to a randomly chosen template sentence in order to generate a possible task description. Given the sent of synonyms and templates, our language generator could create 99,864 unique task descriptions of which we randomly used 45,000 to generate our data set. The final data set contained 22,500 complete task demonstrations composed of the two subtasks (grasping and pouring), resulting in 45,000 training samples. Of these samples, we used 4,000 for validation and 1,000 for testing, leaving 40,000 for training.

**Basic metrics:** Table 1 summarizes the results of testing our model on a set of 100 novel environments. Our model's overall task success describes the percentage of cases in which the cup was first lifted and then successfully poured into the correct bowl. This sequence of steps was successfully executed in $84\%$ of the new environments. Picking alone achieved a $98\%$ success rate, while pouring resulted in $85\%$. We argue that the drop in performance was due to increased linguistic variability when describing the pouring behavior. These results indicate that the model appropriately generalized the trained behavior to changes in object position, verbal command, or perceptual input.

While the task success rate is the most critical metric in such a dynamic control scenario, Table 1 also shows the object detection rate (Dtc), the percentage of dispersed cup content inside the designated bowl (PIn), the percentage of correctly dispersed quantities (QDif), underlining our model's ability to adjust motions based on semantic cues, the mean-average-error of the robot's configuration in radians (MAE), as well as the distance between the robot tool center point and the center of the described target (Dst). The error statistics describe the success rate of the pouring tasks depending on which

Table 2: Generalization to new sentences and changes in illumination

| | Tasks | | | Execution statistics | | |
|---|---|---|---|---|---|---|
| | Pick | Pour | Seq | Dtc | PIn | MAE |
| 1 Illumination | 0.93 | 0.67 | 0.62 | 0.84 | 0.81 | 0.07 |
| 2 Language | 0.93 | 0.69 | 0.64 | 0.86 | 0.83 | 0.09 |
| 3 **Our model** | **0.98** | **0.85** | **0.84** | **0.94** | **0.94** | **0.05** |

combination of visual features was used to uniquely describe the target. For example, when no features were used (column "None"), only one bowl was available in the scene, and no visual features were necessary. Further combinations of color (C), size (S), and shape (F) are outlined in the remaining columns. We noticed that the performance decreased to about $70\%$ when the target bowl needed to be described in terms of both shape and size, even though the individual features had substantially higher success rates of $100\%$ and $85\%$, respectively. It is also notable that our model successfully completed the pouring action in all scenarios in which either the shape or a combination of shape and color were used. The remaining feature combinations reflected the general success rate of $85\%$ for the pouring action.

**Generalization to new users and perturbations:** Subsequently, we evaluated our model's performance when interacting with a new set of four human users, from which we collected 160 new sentences. The corresponding results can be seen in Table 2, row 2. When tested with new language commands, our model successfully performed the entire sequence in $64\%$ of cases. The model nearly doubled the trajectory error rate but maintained a reasonable success rate. It is also observable that

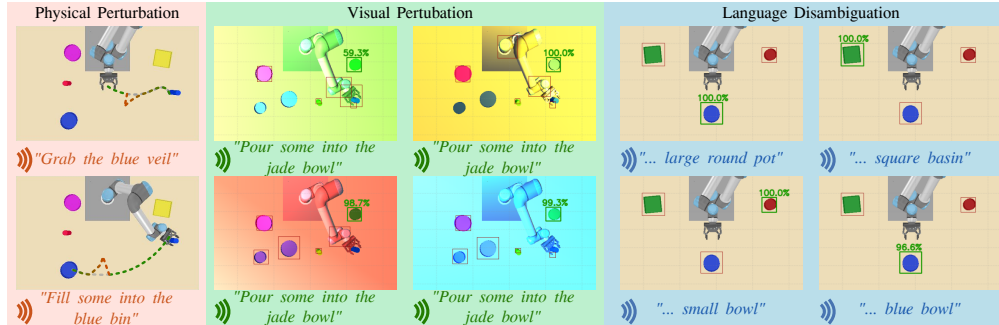

Figure 3: Generalization of our model towards physical perturbations (left), visual perturbations (middle), and verbal disambiguation (right). All experiments used the same model.

most of the failed task sequences primarily resulted from a deterioration in pouring task performance (a reduction from $85\%$ to $69\%$). The picking success rate remained at $93\%$.

Fig. 3 depicts different experiments for testing the ability of our model to deal with physical perturbations, visual perturbations, and linguistic ambiguity. In the physical perturbation experiment, we pushed the robot out of its path by applying a force at around $30\%$ of the distance to the goal. We can see that the robot recovered (red line) from such perturbations in our model. In the visual perturbation experiment (middle), we perturbed the visual appearance of the environment and evaluated if the correct object was detected. We can see that, in all of the above cases, the object was correctly detected at a reasonably high rate (between $59\% - 100\%$). Fig. 3 (right) shows the model's ability to identify the target objects as described in the verbal commands. Depending on the descriptive features used in the task description, the robot assigned probabilities to different objects in the visual field. These values described the likelihood of the corresponding object being the subject of the sentence – a feature that enabled increased transparency of the decision-making process. Fig. 3 (middle) shows examples of the same task executed in differently illuminated scenarios. This experiment highlighted the ability of this approach to cope with perceptual disturbances. Evaluating the model under these conditions yielded a task completion rate of $62\%$ (Table 2, row 1). The main source of accuracy loss was the detection network misclassifying or failing to detect the target object.

**Baselines:** We also compared our approach to two relevant baseline methods. As a first baseline, we evaluated a three-layered LSTM network augmented with extracted features $\mathcal{F}$ of all objects from the object tracker and sentence embedding $s$. The LSTM network concatenated the features in an intermediate embedding and, in turn, predicted the next robot configuration. The second baseline was a current state-of-the-art method called *PayAttention!*, as described in [1]. The objective of PayAttention! was similar to our approach and aimed at converting language and image inputs into low-level robot control policies via imitation learning.

Table 3 compares the results of the two baselines to our model for the pouring, picking, and sequential tasks. Furthermore, the table also shows the percentage of detected objects (Dtc), percentage of dispersed cup content that ended up in the correct bowl (PIn), and the mean-absolute-error (MAE) of the joint trajectory. For fairness, the models were evaluated using two modes: closed loop (CL) and ground truth (GT). In the first mode, using a closed-loop controller, a model was only provided with the start configuration of the robot. In each consecutive time step, the new robot configuration was generated by the simulator after applying the predicted action and calculating dynamics. In the second mode, using ground truth states, a model was constantly provided with the ground truth configurations of the robot as provided by the demonstration. This mode reduced the complexity of the task by eliminating the effect of dynamics and sensor or execution noise but allowed for easier comparison across methods. Results in Table 3 show that the baselines largely failed at executing the sequential task. However, partial success was achieved in the picking task when using the full RNN baseline. Both methods particularly struggled with the more dynamic closed-loop setup, in which they achieved a $0\%$ success rate. Overall, our model (row 5) significantly outperformed both comparison methods. Unlike our model, the PayAttention! method used a fixed alphabet and required information about the task to be extracted from the sentence before use in the model. In contrast, our model could extract a variety of information directly from natural language. We argue that in our case, adverbs and adjectives played a critical role in disambiguating objects and modulating behavior.

PayAttention!, however, primarily focused on objects that could be clearly differentiated by their noun, making it difficult to correctly identify the target objects.

**Ablations of our model**: We studied the influence of auxiliary losses on model performance. Table 1 (rows 1-3) shows the task and execution statistics for different combinations of the auxiliary losses. When training with the trajectory loss (Trj) only, our model successfully completed about $28\%$ of the test cases (row 1). This limited amount of generalization hints at the presence of overfitting. Rather than focusing on task understanding and execution, the network learned to reproduce trajectories. Adding the three remaining controller losses ($W$, $\phi$, and $\Delta_\phi$) aggravated the situation and led to a $0\%$ task completion (row 2). We noticed that attention (Att) was a critical component for training a model with high generalization abilities. Attention ensured that the detected object was in line with the object clause of the verbal task description. A combination of Att and Trj already resulted in a $51\%$ task success rate (row 3). When using the full loss function, including all components, our model achieved an $84\%$ success rate (row 12). This result highlights the critical nature of the loss function, in particular in such a multimodal task. The different objectives related to vision, motion, temporal progression, etc. had to be balanced to achieve the intended generalization.

We also consider an ablation that replaced the attention mechanism with a simple feed forward network. This network took all image features $\mathcal{F}$ as input and generated an intermediate representation via a combination with the sentence embedding $s$ (without any attention mechanism). All other elements of the approach remained untouched. Table 1 (row 4) shows a severe decline in performance when using this modification. This insight underlines the central importance of the attention model in our approach. Pushing the ablation analysis further, we also investigated the impact of the choice of low-level controller. More

Table 3: Comparison to a fundamental baseline and a current state-of-the-art method (PayAttention!) [1].

|  |  | GT | CL | Tasks | | | Execution statistics | | |
|---|---|---|---|---|---|---|---|---|---|
|  |  |  |  | Pick | Pour | Seq | Dtc | PIn | MAE |
| 1 | Full RNN | ✓ |  | 0.58 | 0.00 | 0.00 | 0.52 | 0.07 | 0.30 |
| 2 | Full RNN |  | ✓ | 0.00 | 0.00 | 0.00 | 0.39 | 0.07 | 0.39 |
| 3 | PayAttention! | ✓ |  | 0.23 | 0.08 | 0.00 | 0.66 | 0.41 | 0.13 |
| 4 | PayAttention! |  | ✓ | 0.00 | 0.00 | 0.00 | 0.52 | 0.06 | 0.53 |
| 5 | **Our model** |  | ✓ | **0.98** | **0.85** | **0.84** | **0.94** | **0.94** | **0.05** |

specifically, we evaluated a variant of our model that used attention but replaced the controller module with a three-layer recurrent neural network that directly predicted the next joint configuration (row 5). Again, performance dropped significantly. Finally, we performed an experiment in which we, again, maintained the attention model but replaced only the motor primitives with a feed forward neural network. This variant produced a similar performance to our controller (row 6); the task performance was only marginally lower, by about $5\%$. While this was a reasonable variant of our framework, it lost the ability to generate entire trajectories indicating the robot's future motion. Such lookahead trajectories could be of significant importance for evaluating secondary aspects and safety of a control task (e.g., checking for collision with obstacles, calculating distances to human interaction partners, etc). Therefore, we argue that the specific control model proposed in this paper was more amenable to integration into hierarchical robot control frameworks. Finally, we investigated the impact of the sample size on model performance. Table 1 presents results from different dataset sizes in rows 7 to 11. Significant performance increases could be seen when gradually increasing the sample size from 2,500 to 30,000 training samples. However, the step from 30,000 to 40,000 samples (our main model) only yielded a $4\%$ performance increase, which was negligible compared to the previous increases of $\geq 20\%$ between each step.

# 5 Conclusion

We present an approach for end-to-end imitation learning of robot manipulation policies that combines language, vision, and control. The extracted language-conditioned policies provided a simple and intuitive interface to a human user for providing unstructured commands. This represents a significant departure from existing work on imitation learning and enables a tight coupling of semantic knowledge extraction and control signal generation. Empirically, we showed that our approach significantly outperformed alternative methods, while also generalizing across a variety of experimental setups and achieving credible results on free-form, unconstrained natural-language instructions from previously unseen users. While we use FRCNN for perception and GloVe for language embeddings, our approach is independent of these choices and more recent models for vision and language, such as BERT [13], can easily be used as a replacement.

# 6 Broader impact

Our work describes a machine-learning approach that fundamentally combined language, vision, and motion to produce changes in a physical environment. While each of these three topics has a large, dedicated community working on domain-relevant benchmarks and methodologies, there are only a few works that have addressed the challenge of integration. The presented robot simulation scenario, the experiments, and the presented algorithm[3] provide a reproducible benchmark for investigating the challenges at the intersection of language, vision, and control. Natural language as an input modality is likely to have a substantial impact on how users interact with embedded, automated, and/or autonomous systems. For instance, recent research on the Amazon Alexa [21] suggests that the fluency of the interaction experience is more important to users than the actual interaction output. Surprisingly, "users reported being satisfied with Alexa even when it did not produce sought information"[21].

Beyond the scope of this paper, having the ability to use a natural-language processing system to direct, for example, an autonomous wheelchair [33] may substantially improve the quality of life of many people with disabilities. Natural-language instructions, as discussed in this paper, could open up new application domains for machine learning and robotics, while at the same time improving transparency and reducing technological anxiety. Especially in elder care, there is evidence that interactive robots for physical and social support may substantially improve quality of care, as the average amount of in-person care in only around 24 hours a week. However, for the machine-learning community to enable such applications, it is important that natural-language instructions can be understood across a large number of users, without the need for specific sentence structures or perfect grammar. While far from conclusive, the generalization experiments with free-form instructions from novel human users (see Sec.4) are an essential step in this direction and represent a significant departure from typical evaluation metrics in robotics papers. In particular, we holistically tested whether the translation from verbal description to physical motion in the environment brought about the intended change and task success.

Even before adoption in homes and healthcare facilities, robots with verbal instructions may become an important asset in small and medium-sized enterprises (SMEs). To date, robots have been rarely used outside of heavy manufacturing due to the added burden of complex reprogramming and motion adaptation. In the case of small product batch sizes, as typically used by SMEs, repeated programming becomes economically unsustainable. However, using systems that learn from human demonstration and explanation also comes with the risk of exploitation for nefarious objectives. We mitigated this problem in our work by carefully reviewing all demonstrations, as well as the provided verbal task descriptions, in order to ensure appropriate usage. In addition to the training process, another source of system failure could come from adversarial attacks on our model. This is of particular interest since our model does not only work as software but ultimately controls a physical robotic manipulator that may potentially harm a user in the real world. We addressed this issue in our work by utilizing an attention network that allowed users to verify the selected target object, thereby providing transparency regarding the robot's intended behavior. Despite these features, we argue that more research needs to focus on the challenges posed by adversarial attacks. This statement is particularly true for domains like ours in which machine learning is connected to a physical system that can exert forces in the real world.

## Acknowledgments and Disclosure of Funding

This work was supported by a grant from the Interplanetary Initiative at Arizona State University. We would like to thank Lindy Elkins-Tanton, Katsu Yamane, and Benjamin Kuipers for their valuable insights and feedback during the early stages of this project.

## Footnotes

[1] Subsequently, we would assume, without loss of generality, a seven-degree-of-freedom (DOF) robot – i.e., N = 7.

[2]We use $B = 11$ RBF kernels for each of the 7 DOF with a scale of $\sigma = 0.012$

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
