[Reviews · NeurIPS 2020]

Review 1

Summary and Contributions: The paper introduces a system that incorporates natural-language goal specifications into a learning-from-demonstration system in a simulated robotic pick-and-place task. Its performance compares well with vanilla deep learning networks that are not optimized for the integration task.

Strengths: Integrating multiple domains into machine learning problems is an area of intense research, and is extremely relevant to NeurIPS. The performance reported in the paper is impressive, and it appears to have some ability to generalize within the very narrow task context used in the experiments. The individual components of the system are described well, and clearly show how the system interacts with visual and linguistic data.

Weaknesses: Although the components are described well, their integration is not. The short section 3.4 is all that attempts to explain, and does not go into very much detail. In addition, the simulation environment is very schematic and low-context, and it is not clear that the performance would be reflected in a more sophisticated or complex environment.

Correctness: To a certain extent, yes. The individual descriptions of the construction of the modules make sense, but the integration is less well-described. In addition, it is unclear what the source data for the empirical investigations consist of. The parameters of the experimental scenario are described, but it is difficult to reconcile expert demonstrations and descriptions with the claim of 45,000 training samples. The researchers almost certainly did not elicit 45K demonstrations from humans (if they did, this should be a major claim of the paper!), so it is not clear how ecologically valid their experimental tasks are.

Clarity: The paper is fine on a sentence level, but it is difficult to tease out exactly how the semantic and motor systems are integrated. The explanation is not very clear. Figure 1 helps a little, but a more thorough treatment of how the models interact within the text would be very beneficial to presenting the work in reproducible fashion.

Relation to Prior Work: The prior work discussion is on the short side, but adequate.

Reproducibility: No

Additional Feedback:


Review 2

Summary and Contributions: This paper presents an end-to-end controller for a 7 dof robot arm to manipulate objects conditioned on language and vision inputs. The language considered at training time is only from templates, but evaluations include human language data at inference time. Post-rebuttal: The rebuttal is thoughtful and complete. It provides details that ameliorate all three of my major concerns (human evaluation details, GloVe vs. transformers, and Faster RCNN training). I have updated my score accordingly. It would really strengthen the paper to include especially the details about the human language evaluation in the main body of the paper. Mentioning that Faster RCNN is fine-tuned would also help.

Strengths: Moving end-to-end from language+vision input to control is an emerging area where several research groups are making neat progress, and this paper pushes into that same territory.

Weaknesses: The method requires strong supervision (language to object referents) during training, which requires backing off to templates to get enough training data to use a RL method. It's possible that such strong supervision would better serve a teacher-forced, smaller parameter module with downstream control learned via RL. The main weaknesses of the paper, however, are in the language and vision inputs: The language input is a sequence of lexical GloVe embeddings processed by a GRU, which is fine but feels dated since we know deep contextual models like BERT can generally better process sequences of words than recurrent models on fixed lexical embeddings, and have the bonus of no out-of-vocabulary words at inference time due to WordPiece tokenization. The vision input is a bit bizarre, if I understand it correctly. The rendered scene contains a set of objects from a small number of categories that vary by size and color [Figure 2]. Rather than extract shape and color features from bounding box detections of these objects using lower-level, basic vision algorithms, this method opts for running Faster R-CNN and extracting object features as the detected class plus bounding box edges. Faster R-CNN isn't trained on data that looks anything like this, and the sharp, clear shapes and colors would be much easier to represent with simple color and shape (e.g., FPFH), feature histogram binning.

Correctness: The method seems fine, but the evaluation is in a very toy environment and does not sufficiently discuss the more compelling human evaluation.

Clarity: The method is easy to follow, but the results are presented in some confusing ways, and the annotation data gotten for templates and for test-time evaluation from humans are not well described.

Relation to Prior Work: Missing citation and discussion: Using Natural Language for Reward Shaping in Reinforcement Learning. [https://arxiv.org/pdf/1903.02020.pdf] Prasoon Goyal, Scott Niekum, Raymond J. Mooney. International Joint Conference on Artificial Intelligence (IJCAI), 2019. (A whole survey over similar): A Survey of Reinforcement Learning Informed by Natural Language. [https://arxiv.org/abs/1906.03926] Jelena Luketina, Nantas Nardelli, Gregory Farquhar, Jakob Foerster, Jacob Andreas, Edward Grefenstette, Shimon Whiteson, Tim Rocktäschel. International Joint Conference on Artificial Intelligence (IJCAI), 2019. Might be able to make a stronger argument for using templates if they could be integrated into this formalism for matching test-time utterances to templates seen at training time: Unnatural language processing: bridging the gap between synthetic and natural language data. [https://arxiv.org/abs/2004.13645] Alana Marzoev, Sam Madden, Frans Kaashoek, Mike Cafarella and Jacob Andreas.

Reproducibility: Yes

Additional Feedback: "objects vary in shape, size, color" could mention here that object category varies as well. Line 127 "30000 most used English words" no UNK token needed? Are all the words in the vocabulary already, even from human users for that evaluation? Language templates come from human "experts", but no detail is given about who these experts are, or what the agreement between them as annotators is. Table 2 is confusing to read. Are these "our model" performances under each condition? That's a confusing conclusion to draw since "our model" is written on only one row. Table 3 PayAttention! does so badly it seems like something might be wrong with setup; would be helpful to see training time performance to determine whether this is just a generalization issue. Language disambiguation in Figure 3 is not well explained or shown; it's not clear to me that the model -should- be able to recover from language perturbations, though, since they would change command meaning. I would have liked to see more on "generalization to new users and perturbations". As written, the experiment and result is really skimmed over, with no details about how the langauge and gold end states were gotten. This evaluation could potentially be the strongest in the paper, though. Lines 385-7 make it sound like human eval was done in a physical environment, which I don't think is true. Lines 394-5 says descriptions and trajectories were carefully reviewed? For what? How were they gathered? A lot of information about how the data came to be is missing, especially regarding language. nits: line 93 typo "Such" capitalized


Review 3

Summary and Contributions: The paper presents a framework for learning language-conditioned visuomotor robotic policies from human demonstrations. The framework consists of semantic and control models. Semantic model processes language commands and robot camera images. Language commands are converted into word embeddings using a pre-trained GloVe model and then into sentence embeddings with a GRU cell. Camera images are converted to object features using a Faster R-CNN model and an additional attention layer is learned that combines object features and sentence embeddings. Combined features are then processed by additional networks in the control module to generate parameters of dynamic motor primitives at each time step, which are then executed on the robot. The robot state is tracked with an additional GRU cell. The approach is evaluated in a simulated robotic environment where the robot has to perform pick, place and pouring tasks with objects located at different positions in front of it given language commands. The model is trained on human kinesthetic demonstrations annotated with language commands. The method is shown to outperform previous works and a set of ablations demonstrates the importance of presented design choices.

Strengths: - Introducing language modality into robot learning is important for providing an intuitive way to specify tasks and creates a possibility to generalize to semantic abstractions learned within the language model. - The paper presents a complete language-conditioned robot control system with a range of novel design choices. - Presented approach is evaluated by training on real human data and is shown to outperform previous methods on simulated robotic tasks. Supplementary materials include multiple examples and videos of the robot behavior.

Weaknesses: - Although the paper provides a range of quantitative results for ablations that justify the choice of single framework components, it would interesting to see a more in-depth discussion why simpler components didn’t work well (i.e. replacing motor primitives by feed-forward networks), whether it is connected to a lack of enough data or other aspects of the training setup. - Right now the approach is evaluated on simplistic object shapes, it would be interesting to see evaluation on more articulate objects.

Correctness: Claims and mathematical derivations in the paper are coherent and empirical evaluation correctly shows performance improvements over the baselines.

Clarity: The paper is well-written and easy to understand and follow.

Relation to Prior Work: The paper establishes connection to prior works and uses them as baselines for the experiments.

Reproducibility: Yes

Additional Feedback: Post-rebuttal comments: Thanks for addressing reviewer's comments and providing new details in the rebuttal. Please incorporate these details in the final version of the paper.


Review 4

Summary and Contributions: The paper considers the problem of mapping natural language instructions and initial scene images to robot motion that results in executing the task that is provided through the instruction. This is an important problem to consider as it opens the door to an intuitive interface between robots and humans. The authors propose a network architecture that relies on an attention module linking the instruction to visual observations of the environment, most specifically to target objects. This module produces a single low-level embedding of the scene image and instruction that serves as input to a motor control network which produces the parameters of a relevant motion primitive. The author also allow closed-loop executions through estimating the phase of the motion primitive that they are in. The combined architecture can infer from the instruction and input image what action to do (e.g. pick, poor), how to do it (e.g. pour some, poor all) and where to do it (into a blue bowl). The combined architecture is able to also understand and execute action sequences. The authors motivate their design choices through extensive ablation experiments and comparisons to baselines. I have reviewed the authors response and find that they have sufficiently addressed the raised concerns. Also after discussion with the other reviewers, my rating of the paper has not changed.

Strengths: The paper approaches a hard and important problem in robotics. The experiments are extensive and demonstrate the importance of each ingredients especially for generalising to novel instructions, physical perturbations and changes in illumination. The approach also copes with more complex environments than related work. Very interesting are the translation of sequential tasks and the modulation of the actions (here specifically of the action pour) by words like “little”, “some” and “all”. This would be harder or require more biases if the approach wasn’t end-to-end or if the authors had used some kind of fixed library of skills

Weaknesses: In my opinion the main weakness of the paper is the lack of more quantitive reasoning on why the authors have chosen this end-to-end approach as opposed to a two-stage approach consisting of - a module identifying the target object(s) - a module containing a library of skills (pick, pour) that are trained to perform a task given the target object identified from the first module. For example, for the first module the authors could something similar to Natural Language Object Retrieval  Hu et.al. CVPR 2016. For the second module, the authors could use their demonstrations to learn movement primitives per action from the demonstrations. What I imagine though would be hard is to incorporate any modulation of an action (for example, pour not all but only a little). Something else that could be hard is to deal with sequences of actions. I can see how achieving the sequencing and modulation of an action would require some hand-engineering if going for a two-module approach. But a stronger argument here and a stronger baseline than the Full RNN would bring home the main advantage of this approach in a better way. There is also more work on connecting vision, language and robot motion control that could be discussed and but into relation to this work. See below.

Correctness: The paper seems correct.

Clarity: The paper is very clearly written. Again, the rationale for picking this end-to-end approach could be stronger.

Relation to Prior Work: There’re a few works that I think could be included by the authors and discussed in terms of difference and contribution to their approach. I think one strong difference is that navigation is always simpler in terms of control than manipulation (potentially also something to make more salient in the text). 1.“Learning semantic combinatoriality from the interaction between linguistic and behavioral process”. Sugita et.al. Adaptive behavior, 13(1):33–52, 2005. 2.Language as an abstraction for hierarchical deep reinforcement learning. Jiang et al. In Advances in Neural Information Processing Systems, pages 9414–9426, 2019. 3. Vision-Language Navigation with Self-Supervised Auxiliary Reasoning Task”. Zhu et al. CVPR 2020

Reproducibility: Yes

Additional Feedback: Since the paper is clearly written, the approach seems to be reproducible especially because the authors have also provided code.

[Author Response · NeurIPS 2020]

We thank all reviewers for their constructive feedback! We are encouraged that they found our contribution interesting (**R4**), addressing a hard problem that is important in robotics (**R4**), while also extremely relevant to NeurIPS (**R1**) in the area of robot control from language and vision (**R2**). Our extensive experiments (**R4**) demonstrate impressive performance (**R1**), show the relevance of each component (**R3**, **R4**), and that it is outperforming previous methods on simulated robotic tasks (**R1**, **R3**) in more complex environments (**R4**). Moreover, reviewer **R4** highlighted our model's ability to modulate actions via descriptors like "little" or "all". **In our supplemental material, we provide the full source-code for training and testing, a description and examples of verbal commands by human teachers, and robot videos – please consult index.html**. The primary concern raised regards more detailed explanations of component integration (**R1**), human evaluation (**R2**), FRCNN/Glove choices (**R2**), potential use of simpler methods (**R3**) and modularity (**R4**). We appreciate the feedback and commit to using the extra 1-page allowed for the camera-ready version of accepted papers to substantially expand explanations and address this, as well as add requested references.

**R1,R2 More details on human data collection?** We had five annotators watch pre-recorded agent trajectories and verbally describe the action the robot was performing in unconstrained language. Annotators were graduate students familiar with robotics but not the aim of this project. We collected 200 such descriptions (40 per annotator). Descriptions were transcribed and manually templated by marking replaceable noun-phrases and adverbs. We automatically generated training descriptions by filling these slots with synonyms based on the scene (line 233-239) – **see supplemental material**. This resulted in 99864 unique task descriptions randomly reduced to 45k. In experiment "Generalization to New Users", users typed an instruction and saw the result in a physics-based simulation in real-time (line 263).

**R2 Faster R-CNN isn't trained on data that looks anything like this.** We fine-tuned/pre-trained the FRCNN model (from ResNet-101 trained on COCO) on 40k arbitrarily generated environments of our simulator. After fine-tuning, the certainty on FRCNN on our objects is above 98%. FRCNN was chosen because it is a commonly-used method with reasonably high performance and largely understood pros and cons. We do not claim contributions to object detection, but rather to the integration of language and vision for robot control. While we are open to using simpler approaches, FPFH would not be applicable since it is a 3D point-cloud approach requiring access to a depth camera.

**R1**, **R3 How are all components integrated? Simpler components?** The components of our model, explained in section 3.2 and 3.3, are integrated sequentially. After pre-processing of the image and language data (section 3.1), our model takes the input data and converts them into a task-specific embedding within the semantic model (section 3.2). In turn, this embedding is then used to generate the hyper-parameters of the low-level controller (line 152), namely weights, current phase, and desired phase progression (line 173). With these parameters, we define a motor primitive determining the next robot motion (line 186) as well as the entire trajectory. Especially the latter is a benefit of our approach over simpler FF networks. We commit to substantially expanding section 3.4 (**R1**) in the camera-ready version as well as adding a discussion explaining why simpler models did not work as well (**R3**).

**R2 Why is PayAttention performing poorly?** We used the original code from the PayAttention paper and have repeatedly sought out the assistance of the main author (who thankfully provided substantial support) to ensure that our usage exactly follows the method and protocol required. Regarding performance, we argue that in our case, adverbs and adjectives play a critical role in disambiguating objects and modulating the behavior. PayAttention, however, primarily focuses on objects that can be clearly differentiated by their noun.

**R4 Why end-to-end and not a multi-staged approach?** As the reviewer pointed out, multi-staging requires a significant amount of additional (human) feature engineering, which would, at the same time, limit our approach to these features and may also be fragile in terms of generalizing to new words. Our framework learns how language affects the behavior (type, goal position, velocity, etc.) automatically, while also learning the control itself. Another advantage of end-to-end is that the overall system can be trained such that the individual components harmonize. This is particularly important for the interplay of language embedding and control when using language as a modifier for trajectory behaviors.

**R2 We know deep contextual models like BERT can generally better process sequences.** This is a great suggestion; our approach easily allows for the integration of alternative approaches for language, e.g., BERT. GloVe was chosen due to its simplicity, but to demonstrate our framework's extensibility, we incorporated BERT as suggested. Due to time constraints, we trained only a single model **which achieved 97% on the picking and 65% on the pouring task**, which is comparable to our original model with 98% and 85% respectively. Better performance can likely be achieved with more careful integration and tuning of the parameters. Still, we argue that this ability to replace language models with more recent SotA methods quickly is an appealing feature of our approach that allows steady improvements.

**R2 The method requires strong supervision (language to object referents) during training, which requires backing off to templates to get enough training data to use an RL method.** We would like to clarify: our approach is a pure imitation learning technique, and no RL is involved. RL could potentially be an option, but learning policies that bridge language, vision, and control with a limited amount of trials (< 100K) would be extremely challenging.

[Meta-Review · NeurIPS 2020]

The reviewers initially had concerns, especially related to feature representations. lack of details about the human evaluation, and the justification for the end to end approach. However, the reviewers agreed that the author response was well written and addressed any major concerns about the paper. There was still a sentiment that integration of the ideas could be stronger, and that more complex, realistic environments would improve the paper, but that it was strong enough to be accepted as-is (though the authors are encouraged to take the advice of the reviewers for the camera ready).